# Catastrophic Forgetting Meets Negative Transfer: Batch Spectral Shrinkage for Safe Transfer Learning

**Xinyang Chen,**[*] **Sinan Wang,**[*] **Bo Fu, Mingsheng Long** (✉),[†] **and Jianmin Wang**
School of Software, BNRist, Tsinghua University, China
Research Center for Big Data, Tsinghua University, China
National Engineering Laboratory for Big Data Software
{chenxiny17,wang-sn17}@mails.tsinghua.edu.cn, {mingsheng,jimwang}@tsinghua.edu.cn

## Abstract

Before sufficient training data is available, fine-tuning neural networks pre-trained on large-scale datasets substantially outperforms training from random initialization. However, fine-tuning methods suffer from a dilemma across *catastrophic forgetting* and *negative transfer*. While several methods with explicit attempts to overcome catastrophic forgetting have been proposed, negative transfer is rarely delved into. In this paper, we launch an in-depth empirical investigation into negative transfer in fine-tuning and find that, for the weight parameters and feature representations, transferability of their spectral components is diverse. For *safe* transfer learning, we present **Batch Spectral Shrinkage (BSS)**, a novel regularization approach to penalizing smaller singular values so that untransferable spectral components are suppressed. BSS is orthogonal to existing fine-tuning methods and is readily pluggable into them. Experimental results show that BSS can significantly enhance the performance of state-of-the-art methods, especially in few training data regime.

## 1 Introduction

Deep learning has made revolutionary changes to diverse machine learning problems and applications. During the past few years, significant improvements on various tasks have been achieved by deep neural networks [17, 33, 10, 35]. However, training deep neural networks from scratch is time-consuming and laborious, and the excellent performance of such deep neural networks depends on large-scale labeled datasets which we may have no access to in many practical scenarios.

Fortunately, deep feature representations learned on large-scale datasets are transferable across several tasks and domains [25, 7, 45]. Thus, *fine-tuning*, a simple yet effective method that exploits this nice property of deep representations, is widely adopted, especially before sufficient training data is available [9]. Under this well-established paradigm, deep neural networks are firstly pre-trained on large-scale datasets and then fine-tuned to target tasks, requiring relatively smaller training samples.

To a certain extent, fine-tuning alleviates deep neural networks' hunger for data. However, adequate amount of training data for target tasks is still a prerequisite for the effectiveness of vanilla fine-tuning methods. When the requirement of training data cannot be satisfied, two hidden issues of fine-tuning will become extremely severe, seriously hampering the generalization performance of deep models. The first is **catastrophic forgetting** [14], which is the tendency of the model to lose previous learnt knowledge abruptly while it may incorporate information relevant to target tasks, leading to overfitting. The second is **negative transfer** [37]. Not all pre-trained knowledge is transferable across domains, and an indiscriminate transfer of all knowledge is detrimental to the model.

---

[*]Authors contributed equally

[†]Corresponding author: Mingsheng Long (mingsheng@tsinghua.edu.cn)

Table 1: Comparison of Different Fine-Tuning Methods (—: unknown)

| Method | Target Dataset Size | | | Technical Challenge | |
|---|---|---|---|---|---|
| | large | medium | small | catastrophic forgetting | negative transfer |
| $L^2$ | ✓ | — | ✗ | ✗ | ✗ |
| $L^2$-SP [20] | ✓ | — | ✗ | ✓ | ✗ |
| DELTA [19] | ✓ | — | ✗ | ✓ | ✗ |
| **BSS (Proposed)** | ✓ | ✓ | ✓ | ✗ | ✓ |

Incremental learning [30, 18, 21, 32, 44] extends the existing model's knowledge continuously with gradually available training data. Various measures have been taken to curb the tendency of forgetting previously learnt knowledge while acquiring new knowledge. Note that the original motivation of mitigating *catastrophic forgetting* for incremental learning and fine-tuning is quite different. In the context of incremental learning, the model performance on both old and new tasks makes sense, while when it comes to fine-tuning, only target tasks are concerned. In this paper, catastrophic forgetting refers specifically to forgetting the pre-trained knowledge beneficial to target tasks. During the past few years, a few transfer learning penalties [20, 19] have been proposed to constrain parameters on maintaining pre-trained knowledge. Specially, $L^2$-SP [20] considers that weight parameters should be driven to pre-trained values instead of the origin and takes the advantage of all pre-trained weights to refrain networks from forgetting useful information. DELTA [19] utilizes discriminative knowledge in feature maps and imposes feature map regularization by the attention mechanism.

Methods above largely alleviate the problem of catastrophic forgetting by drawing weight parameters close to pre-trained values or aligning transferable channels in feature maps. Still, *negative transfer* has not been attached with enough importance and is often overlooked in deep methods. However, when the amount of training examples on the target domain is limited, overly retaining pre-trained knowledge will deteriorate target performance and negative transfer will become prominent. It is thereby apparent that catastrophic forgetting and negative transfer constitute a *dilemma*, which should be solved jointly for *safe* transfer learning. In this paper, we explore fine-tuning against negative transfer and propose a novel regularization approach to restraining detrimental pre-trained knowledge during fine-tuning. A comparison of these fine-tuning methods is presented in Table 1.

Based on Singular Value Decomposition (SVD), we investigate which spectral components of weight parameters and feature representations are untransferable across domains, and make two observations. For weight parameters, in high layers, the spectral components with small singular values are not transferable. For feature representations, an interesting finding is that with sufficient training data, the spectral components with small singular values are decayed autonomously during fine-tuning. Inspired by this inherent mechanism, we propose **Batch Spectral Shrinkage (BSS)**, a general approach to inhibiting negative transfer by suppressing the spectral components with small singular values that correspond to detrimental pre-trained knowledge. BSS is orthogonal to existing methods for mitigating catastrophic forgetting, and can be easily embedded into them to tackle the dilemma. Experiments confirm the effectiveness of BSS in mitigating negative transfer, especially when the amount of available training data is limited, yielding state-of-the-art results on several benchmarks.

## 2 Related Work

Transfer learning, an important machine learning paradigm, is committed to transferring knowledge obtained on a source domain to a target domain [2, 26]. There are several different scenarios of transfer learning, such as domain adaptation [31] and multi-task learning [2], while inductive transfer learning is the most practical one. In inductive transfer learning, 1) the target task is different from the source task (different label spaces), and 2) there is labeled data in the target domain.

Fine-tuning is the de facto approach to inductive transfer of deep models, where we have a pre-trained model from the source domain but have no access to the source data. To utilize pre-trained knowledge obtained on the source domain, Donahue *et al.* [7] employed a label predictor to classify features extracted by the pre-trained model. This method directly reused a substantial part of the weight parameters, which inhibits catastrophic forgetting (relevant information eliminated) but exacerbates the risk of negative transfer (irrelevant information retained). Later, deep networks proved to be able

to learn transferable representations [45]. To explore potential factors affecting deep transfer learning performance, Huh *et al.*[12] empirically analyzed features extracted by various networks pre-trained on ImageNet. Recently, numerous approaches were proposed to advance this field, including filter distribution constraining [1], sparse transfer [22], and filter subset selection [8, 4]. Further, Simon *et al.* [15] empirically studied what factors impact inductive transfer of deep models.

Catastrophic forgetting is an inevitable problem of incremental learning or lifelong learning [36]. To overcome this limitation, incremental moment matching [18] and "hard attention to the task" [32] have been proposed. In inductive transfer learning, the pre-trained networks also have the tendency to lose previous learnt knowledge abruptly while incorporating information relevant to target tasks. By driving weight parameters to initial pre-trained values, $L^2$-SP [20] enhances model performance for target tasks while avoiding degradation in accuracy on pre-trained datasets. Inspired by knowledge distillation for model compression [29, 11, 46, 43], Li *et al.* [19] proposed the idea of "unactivated channel re-usage" and presented DELTA, a feature map regularization with attention.

Above methods have achieved remarkable performance gains and alleviated catastrophic forgetting to varying degrees. However, negative transfer, a major challenge in domain adaptation [31, 34, 38, 40, 41], has rarely been considered in inductive transfer learning. In this paper, from the perspective of inhibiting negative transfer during fine-tuning, we propose Batch Spectral Shrinkage (BSS), a novel regularization approach orthogonal to existing methods, to enhance fine-tuned models' performance.

## 3  Catastrophic Forgetting Meets Negative Transfer

In inductive transfer learning (fine-tuning), we have access to a target domain with $n$ labeled examples and a network pre-trained on a source domain. Different from domain adaptation [26], in fine-tuning the source domain is *inaccessible* at training. For classification tasks, typically, the network consists of two parts: the shared sub-network (feature extractor $F$) and the task-specific architecture (classifier $G$). We denote by $F^0$ and $G^0$ the corresponding parts with pre-trained weights respectively.

There are two potential pitfalls inductive transfer learning may have. The first one is **catastrophic forgetting**, which refers to a tendency of the model to abruptly forget previously learnt knowledge upon acquiring new knowledge. The second is **negative transfer**, a process where the model transfers knowledge irrelevant to target tasks, and leads to negative impacts on model performance. Almost all existing deep methods concentrate on the former. It is natural to raise the following questions: 1) Does negative transfer really exist in fine-tuning? 2) If it does, how does it affect model performance?

### 3.1  Regularizations for Transfer Learning

We first review existing inductive transfer learning methods. Almost all fine-tuning methods can be formulated as follows:

$$\min_{\mathbf{W}} \sum_{i=1}^{n} L(G(F(\mathbf{x}_i)), y_i) + \Omega(\cdot), \tag{1}$$

where $\mathbf{W}$ refers to the weight parameters of models, $L(\cdot, \cdot)$ denotes the loss function and $\Omega(\cdot)$ is the regularization term on the weights or on the features extracted by the model. Next we will discuss three fine-tuning penalties and their corresponding effects on mitigating catastrophic forgetting.

**$L^2$ penalty.**  The common penalty for transfer learning is $L^2$ penalty, also known as weight decay:

$$\Omega(\mathbf{W}) = \frac{\alpha}{2} \left\| \mathbf{W} \right\|_2^2, \tag{2}$$

where $\alpha$ is a hyperparameter to control the strength of this regularization term. $L^2$ penalty tries to drive the network parameters to zero, without considering catastrophic forgetting or negative transfer.

**$L^2$-SP.**  The key concept of $L^2$-SP penalty [19] is "starting point as reference":

$$\Omega(\mathbf{W}) = \Omega(\mathbf{W}, \mathbf{W}^0) = \frac{\beta}{2} \left\| \mathbf{W}_S - \mathbf{W}_S^0 \right\|_2^2 + \frac{\alpha}{2} \left\| \mathbf{W}_{\overline{S}} \right\|_2^2, \tag{3}$$

where $\mathbf{W}_S^0$ is the pre-trained weight parameters of the shared architecture (feature extractor $F_0$), $\mathbf{W}_S$ is weight parameters of $F$, $\mathbf{W}_{\overline{S}}$ is weight parameters of the task-specific classifier $G$, $\beta$ is a trade-off

hyperparameter to control the strength of the penalty. $L^2$-SP penalty tries to drive weight parameters to pre-trained values. Xuhong *et al.* [20] empirically proved that $L^2$-SP reduces drop in accuracy of networks on source tasks after fine-tuning, revealing that $L^2$-SP can alleviate catastrophic forgetting.

**DELTA.** Based on the key insight of "unactivated channel re-usage", Li *et al.* [19] proposed a regularized transfer learning framework, DELTA. Specifically, DELTA selects the discriminative features from higher layer outputs with a supervised attention mechanism. $\Omega(\mathbf{W})$ is formulated as:

$$\Omega(\mathbf{W}) = \Omega(\mathbf{W}, \mathbf{W}^0, \mathbf{x}_i, y_i, z) = \gamma \cdot \Omega'(\mathbf{W}, \mathbf{W}^0, \mathbf{x}_i, y_i, z) + \kappa \cdot \Omega''(\mathbf{W} \backslash \mathbf{W}^0)$$

$$\Omega'(\mathbf{W}, \mathbf{W}^0, \mathbf{x}_i, y_i, z) = \sum_{j=1}^{N} \mathrm{D}_j(z, \mathbf{W}^0, \mathbf{x}_i, y_i) \cdot \left\| \mathrm{FM}_j(z, \mathbf{W}, \mathbf{x}_i) - \mathrm{FM}_j(z, \mathbf{W}^0, \mathbf{x}_i) \right\|_2^2 \quad (4)$$

$$\mathrm{D}_j(z, \mathbf{W}^0, \mathbf{x}_i, y_i) = \mathrm{softmax}(L(z(\mathbf{x}_i, \mathbf{W}^{0 \backslash j}), y_i) - L(z(\mathbf{x}_i, \mathbf{W}^0), y_i))$$

where $z$ is the model, $\Omega'$ is behavioral regularizer, $\Omega''$ constrains the $L^2$-norm of the private parameters in $\mathbf{W}$; $\mathrm{D}_j(z, \mathbf{W}^0, \mathbf{x}_i, y_i)$ refers to the behavioral difference between the two feature maps (FM) and the weight assigned to the $j$th filter and the $i$th image (for $1 < j < N$); $\gamma$ and $\kappa$ are trade-off hyperparameters to control the strength of the two regularization terms. DELTA alleviates catastrophic forgetting by aligning the behaviors of certain higher layers of the target network to the source one.

## 3.2 Negative Transfer in Fine-tuning

In this section, we will investigate whether negative transfer exists and whether it has a negative impact on the model's performance. We design an experiment based on $L^2$ penalty and $L^2$-SP penalty. ResNet-50 [10] pre-trained on ImageNet is chosen as the backbone and MIT Indoors 67 [28] is the target dataset. The training details are consistent with Section 5. We sample the training datasets at the rates of $15\%$, $30\%$, $50\%$ and $100\%$ to construct new training datasets of different sizes.

Contrary to what one might suppose, as shown in Figure 1(a), $L^2$-SP penalty worsens the model's performance, compared with $L^2$ penalty, especially when the amount of training data is limited. $L^2$-SP penalty explicitly promotes the similarity of the final solution with the initial model to alleviate catastrophic forgetting, while $L^2$ does not. Although only the behaviors of certain higher layers of the target network are aligned to the source one, $L^2$-SP still aggravates negative transfer, in that the pre-trained knowledge irrelevant to the target tasks is still transferred forcefully.

As negative transfer does exist, further, we want to answer two questions: 1) **Which** part of weight parameters and feature representations causes negative transfer? 2) **How** to mitigate this problem?

## 3.3 Why Negative Transfer?

In this section, we will explore which part of the weight parameters $\mathbf{W}$ and feature representations $\mathbf{f} = F(\mathbf{x})$ may not be transferable and may negatively impact the model accuracy. ResNet-50 [10] pre-trained on ImageNet is chosen as the backbone and MIT Indoors 67 is the target dataset. Weight parameters and feature representations of both pre-trained and fine-tuned networks are analyzed.

**Corresponding Angle.** Principal angles [24] have been introduced to measure the similarity of subspaces. However, it is unreasonable to calculate the principal angles by completing the pairing between whole eigenvectors in subspaces with the smallest angle, regardless of their relative singular values, because eigenvectors with large singular values and small singular values have different roles in matrices. Inspired by [3], we use corresponding angles, denoted by $\theta$. Definitions are as follows:

**Definition 1 (Corresponding Angle)** *It is the angle between two eigenvectors which are equally important in their matrices. That is, they are related to the same index in the singular value matrices.*

The cosine value of the corresponding angle is calculated as

$$\cos(\theta_i) = \frac{\langle \mathbf{u}_{1,i}, \mathbf{u}_{2,i} \rangle}{\|\mathbf{u}_{1,i}\| \|\mathbf{u}_{2,i}\|}, \quad (5)$$

where $\mathbf{u}_{1,i}$ is the $i$th eigenvector with the $i$th largest singular value in one matrix, and similarly for $\mathbf{u}_{2,i}$ in another matrix. We will use $\theta$ to measure the transferability of eigenvectors in weight matrices. Intuitively, eigenvectors with smaller corresponding angle across domains imply better transferability.

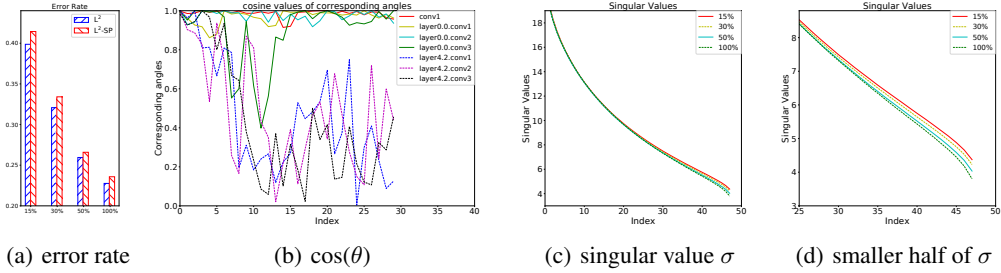

| (a) error rate | (b) $\cos(\theta)$ | (c) singular value $\sigma$ | (d) smaller half of $\sigma$ |

Figure 1: Analysis of negative transfer: (a) Error rates of fine-tuned models with $L^2$ and $L^2$-SP penalties; (b) Cosine values of the corresponding angles between $\mathbf{W}$ and $\mathbf{W}^0$; (c) All singular values of feature matrices extracted on four configurations for the dataset MIT Indoors 67, with random sampling rates $15\%$, $30\%$, $50\%$ and $100\%$ respectively; (d) The smaller half of singular values in (c).

**Weights.** We denote by $\mathbf{W}^0$ and $\mathbf{W}$ the pre-trained weight parameters of ResNet-50 on ImageNet and fine-tuned weight parameters on MIT Indoors 67 respectively. For a conv2d layer, its parameters form a four-dimensional tensor with the shape of $(c_{i+1}, c_i, k_h, k_w)$. We unfold this tensor to a matrix with the shape $(c_{i+1}, c_i \cdot k_h \cdot k_w)$ and perform SVD to obtain eigenvectors $\mathbf{U}$ and singular values $\mathbf{\Sigma}$:

$$\mathbf{W} = \mathbf{U}\mathbf{\Sigma}\mathbf{V}^{\mathsf{T}}. \tag{6}$$

Then, following Equation (5), relative angles $\theta$ are calculated in every layers between $\mathbf{W}$ and $\mathbf{W}^0$. Corresponding angles in four lower layers (the first convolutional layer and three convolutional layers in the first residual block) and three higher layers (three convolutional layers in the last residual block) are shown in Figure 1(b), the former with solid lines and the latter with dotted lines. We can observe that for the lower layers, eigenvectors in $\mathbf{W}$ and $\mathbf{W}^0$ have small relative angles, which means these weight parameters are transferable. However, in the higher layers, only eigenvectors corresponding to relatively larger singular values have small corresponding angles. So aligning all weight parameters indiscriminately to the initial pre-trained values is risky to negative transfer.

**Features.** Analyzing feature representations, rather than weight parameters, is more straightforward. We will analyze the characteristics of feature representations produced by models with different generalization performance. As the size of training dataset has a profound impact on model performance, we sample MIT Indoors 67 at the rates of $15\%$, $30\%$, $50\%$ and $100\%$ to construct new training datasets. We fine-tune ImageNet pre-trained ResNet-50 on these four datasets and then obtain four models.

The feature extractor fine-tuned on target datasets is denoted by $F$ and the feature vector is calculated by $\mathbf{f}_i = F(\mathbf{x}_i)$. Every feature matrix $\mathbf{F} = [\mathbf{f}_1 \ldots \mathbf{f}_b]$ is composed of a batch size $b$ of feature vectors. Again, we apply SVD to compute all singular eigenvectors $\mathbf{U}$ and values $\mathbf{\Sigma}$ of the feature matrices:

$$\mathbf{F} = \mathbf{U}\mathbf{\Sigma}\mathbf{V}^{\mathsf{T}}. \tag{7}$$

The main diagonal elements $[\sigma_1, \sigma_2..., \sigma_b]$ of the singular value matrix $\mathbf{\Sigma}$ (a rectangular diagonal matrix) are drawn in Figure 1(c) and Figure 1(d) in descending order, measuring the importance of eigenvectors. Figure 1(c) contains all these singular values, and Figure 1(d) contains the smaller half of them. As justified by [9], with sufficient labeled data, fine-tuning and training from scratch achieve comparably best results. Hence models fine-tuned on larger datasets can have stronger generalization performance. It is important to observe that the relatively small singular values of features extracted by such models are suppressed significantly, indicating that the spectral components corresponding to relatively small singular values are relevant to the variation of training data that are less transferable. Consequently, promoting the similarity between these components will give rise to negative transfer.

## 4 Approach

We stress that catastrophic forgetting and negative transfer are equally important and constitute an inherent dilemma for fine-tuning. While the previous section focuses on why negative transfer occurs, this section presents how to alleviate negative transfer without casting aside pre-trained knowledge.

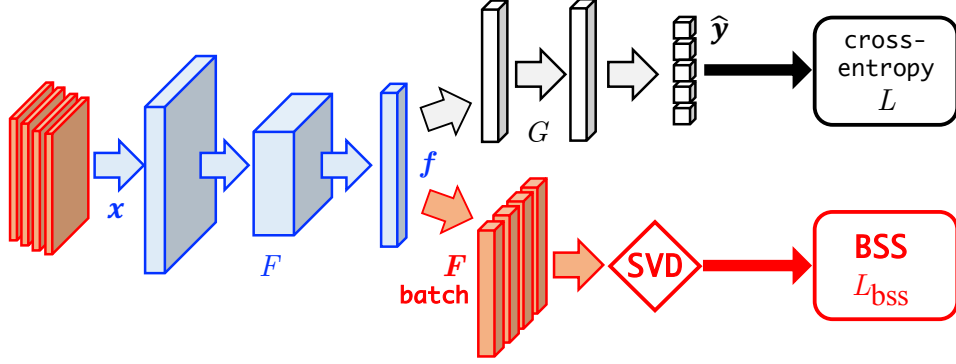

Figure 2: The architecture of Batch Spectral Shrinkage (**BSS**). **BSS** is a new regularization approach to overcoming negative transfer in fine-tuning, which is readily pluggable into existing methods and is end-to-end trainable with **differentiable SVD** natively supported in PyTorch (best viewed in color).

The analysis above shows that both weight parameters and feature representations are partially transferable. For weight parameters, almost all eigenvectors in lower layers are transferable, while in higher layers only eigenvectors with large singular values are transferable. For feature representations, an expanded dataset can enhance the performance of models and suppress the eigenvectors with small singular values of the feature matrices. This inspires us to suppress the importance of spectral components that are untransferable, especially when the number of training data examples is limited. As applying SVD to high-dimensional weight matrices is extremely costly, for untransferable layers with huge weight parameters, we perform spectral component shrinkage on the feature matrices only.

## 4.1 Batch Spectral Shrinkage

The above decomposition analysis of feature matrices brings us the key inspiration. We propose a new regularization approach, **Batch Spectral Shrinkage (BSS)**, to restrain negative transfer during fine-tuning through directly suppressing the small singular values of the feature matrices. Detailed procedures are as follows: 1) Constructing a feature matrix $\mathbf{F}$ from a batch size $b$ of feature vectors $\mathbf{f}$; 2) Applying SVD to compute all singular values of $\mathbf{F}$ as Equation (7); 3) Penalizing the smallest $k$ singular values $[\sigma_1, \sigma_2..., \sigma_b]$ in the diagonal of singular value matrix $\mathbf{\Sigma}$ to mitigate negative transfer:

$$L_{\text{bss}}(F) = \eta \sum_{i=1}^{k} \sigma_{-i}^2,$$ (8)

where $\eta$ is a trade-off hyperparameter to control the strength of spectral shrinkage, $k$ is the number of singular values to be penalized, and $\sigma_{-i}$ refers to the $i$-th smallest singular value.

**Computational Complexity.** For a $p \times q$ matrix, the time complexity of full SVD that computes all singular values is $O(\min(p^2 q, pq^2))$. The time cost of performing SVD on a nearly squared matrix is unacceptable, e.g. weight matrices of deep networks. The complexity of BSS is $O(b^2 d)$ where $d$ is the dimension of features and $b$ is the batch size. Typically, as $b$ is relatively small, say $b = 48$, the overall computational budget of BSS is nearly negligible in fine-tuning through the mini-batch SGD.

## 4.2 Models with Batch Spectral Shrinkage

Almost all of existing fine-tuning methods concentrate on catastrophic forgetting. BSS, as a novel regularization approach we propose from another perspective, boosts fine-tuning through inhibiting negative transfer, making itself orthogonal to previous methods. BSS is lightweight and pluggable readily into existing fine-tuning methods, e.g. $\text{L}^2$, $\text{L}^2$-SP [20] and DELTA [19]. Figure 2 showcases the architecture of $\text{L}^2$+BSS. BSS embedded into existing fine-tuning scenarios can be formulated as:

$$\min_{\mathbf{W}} \sum_{i=1}^{n} L(G(F(\mathbf{x}_i)), y_i) + \Omega(\mathbf{W}) + L_{\text{bss}}(F).$$ (9)

# 5 Experiments

We embed BSS into representative inductive transfer learning methods mentioned above, including $L^2$, $L^2$-SP and DELTA, and evaluate these methods on several visual recognition benchmarks. Except that, BSS is also explored in other scenarios, such as incremental learning and natural language processing. Code and datasets are available at github.com/thuml/Batch-Spectral-Shrinkage.

## 5.1 Setup

**Stanford Dogs** [13] contains 20,580 images of 120 breeds of dogs from around the world. Each category is composed of exactly 100 training examples and around 72 testing examples.

**Oxford-IIIT Pet** [27] is a 37-category pet dataset with roughly 200 images for each class.

**CUB-200-2011** [42] is a dataset for fine-grained visual recognition with 11,788 images in 200 bird species. It is an extended version of the CUB-200 dataset, roughly doubling the number of images.

**Stanford Cars** [16] contains 16,185 images of 196 classes of cars. Each category has been split roughly in a 50-50 split. There are 8,144 images for training and 8,041 images for testing.

**FGVC Aircraft** [23] is a benchmark for the fine-grained visual categorization of aircraft. The dataset contains 10,200 images of aircraft, with 100 images for each of the 102 different aircraft variants.

To explore the impact of *negative transfer* with different numbers of training examples, we create four configurations for each dataset, which respectively have 15%, 30%, 50%, and 100% randomly sampled training examples for each category. Following the previous protocols [20, 19], we employ ResNet-50 [10] pre-trained on ImageNet [5] as the source model. The last fully connected layer is trained from scratch, with learning rate set to be 10 times those of the fine-tuned layers, which is a *de facto* configuration in fine-tuning. We adopt mini-batch SGD with momentum of 0.95 using the progressive training strategies in [20] except that the initial learning rate for the last layer is set to 0.01 or 0.001, depending on the tasks. We set batch size to 48. In all experiments with BSS, the trade-off hyperparameter $\eta$ is fixed to 0.001 and $k$ is set to 1. Each experiment is repeated five times, and the average top-1 accuracy is reported in Table 2.

## 5.2 Results and Analyses

**Results.** The top-1 classification accuracies are shown in Table 2. It is observed that BSS produces boosts in accuracy with fewer training data for most methods on most datasets. However, performance gains on Stanford Dogs and Oxford-IIIT Pet are not very obvious, indicating that the transferability of pre-trained knowledge across these datasets plays a major role and thus negative transfer impact is not as serious as expected. Embedding BSS into $L^2$-SP and DELTA, $L^2$-SP+BSS and DELTA+BSS alleviate negative transfer and catastrophic forgetting simultaneously to yield state-of-the-art results.

**Negative Transfer.** To delve into BSS, we remove the spectral components corresponding to the smallest $r$ singular values, named Truncated SVD (**TSVD**). Formally, SVD is performed on mini-batch feature matrix $\mathbf{F}$, yielding $b$ singular vectors and values. Then only the $b - r$ column vectors of $\mathbf{U}$ and $b - r$ row vectors of $\mathbf{V}^\mathsf{T}$ corresponding to the $b - r$ largest singular values $\boldsymbol{\Sigma}_{b-r}$ are calculated. Finally, the rest of the matrix $\mathbf{F}$ is discarded, with an approximate feature matrix $\mathbf{F}_{b-r}$ reconstructed:

$$\mathbf{F} = \mathbf{U}\boldsymbol{\Sigma}\mathbf{V}^\mathsf{T}, \quad \mathbf{F}_{b-r} = \mathbf{U}_{b-r}\boldsymbol{\Sigma}_{b-r}\mathbf{V}_{b-r}^\mathsf{T}. \tag{10}$$

ResNet-50 pre-trained on ImageNet is employed as the base model and Stanford Dogs is the target dataset. We analyze the performance of TSVD ($r = 1, 2, 4, 8$) with $L^2$ penalty. Results are shown in Figure 3(a). We find that when the dataset is relatively small, TSVD with a larger $r$ leads to better performance, which proves that spectral components corresponding to relatively small singular values have negative impact on transfer learning. Thus, BSS is a reasonable approach to inhibiting negative transfer. However, when sufficient training data is available, a larger $r$ may deteriorate the accuracy.

**Singular Values.** Singular values of features extracted by the networks fine-tuned with regularization $L^2$+BSS and $L^2$ are shown in Figure 3(b)–3(c). The former is with dotted line and the latter is with solid line. Although $k$ in Equation (8) is set to 1, more than one singular values are suppressed, indicating that feature matrices are capable of automatically adjusting singular value distributions. $k = 1$ is adequate for most cases, and a larger $k$ may display equal effect with a larger trade-off hyperparameter $\eta$. BSS is effective in suppressing small singular values to combat negative transfer.

Table 2: Comparison of Top-1 Accuracy with Different Methods (Backbone: ResNet-50)

| Dataset | Method | Sampling Rates | | | |
|---|---|---|---|---|---|
| | | 15% | 30% | 50% | 100% |
| Stanford Dogs | $L^2$ | 81.05±0.18 | 84.47±0.23 | 85.69±0.21 | 86.89±0.32 |
| | **$L^2$+BSS** | 81.86±0.19 | 84.79±0.18 | 86.00±0.22 | **87.18**±0.14 |
| | $L^2$-SP [20] | 81.41±0.23 | 84.88±0.15 | 85.99±0.18 | 86.72±0.20 |
| | **$L^2$-SP+BSS** | **82.20**±0.27 | **85.06**±0.17 | **86.18**±0.05 | 86.91±0.19 |
| | DELTA [19] | 81.46±0.18 | 83.66±0.29 | 84.73±0.16 | 86.01±0.22 |
| | **DELTA+BSS** | 81.93±0.29 | 84.33±0.16 | 85.30±0.30 | 86.54±0.14 |
| CUB-200-2011 | $L^2$ | 45.25±0.12 | 59.68±0.21 | 70.12±0.29 | 78.01±0.16 |
| | **$L^2$+BSS** | 47.74±0.23 | **63.38**±0.29 | **72.56**±0.17 | 78.85±0.31 |
| | $L^2$-SP [20] | 45.08±0.19 | 57.78±0.24 | 69.47±0.29 | 78.44±0.17 |
| | **$L^2$-SP+BSS** | 46.77±0.19 | 60.89±0.28 | 72.33±0.26 | **79.36**±0.12 |
| | DELTA [19] | 46.83±0.21 | 60.37±0.25 | 71.38±0.20 | 78.63±0.18 |
| | **DELTA+BSS** | **49.77**±0.07 | 62.95±0.18 | 72.31±0.38 | 79.02±0.21 |
| Stanford Cars | $L^2$ | 36.77±0.12 | 60.63±0.18 | 75.10±0.21 | 87.20±0.19 |
| | **$L^2$+BSS** | 40.57±0.12 | 64.13±0.18 | 76.78±0.21 | **87.63**±0.27 |
| | $L^2$-SP [20] | 36.10±0.30 | 60.30±0.28 | 75.48±0.22 | 86.58±0.26 |
| | **$L^2$-SP+BSS** | 39.44±0.18 | 64.41±0.19 | 76.56±0.28 | 87.38±0.23 |
| | DELTA [19] | 39.37±0.34 | 63.28±0.27 | 76.53±0.24 | 86.32±0.20 |
| | **DELTA+BSS** | **41.92**±0.16 | **64.67**±0.28 | **77.58**±0.33 | 86.32±0.25 |
| Oxford-IIIT Pet | $L^2$ | 86.56±0.21 | 89.99±0.35 | 91.22±0.19 | 92.75±0.25 |
| | **$L^2$+BSS** | **87.57**±0.13 | **90.46**±0.21 | **92.07**±0.29 | **93.30**±0.14 |
| | $L^2$-SP [20] | 86.78±0.21 | 90.00±0.23 | 90.65±0.18 | 92.29±0.22 |
| | **$L^2$-SP+BSS** | 87.53±0.36 | 90.13±0.21 | 91.03±0.09 | 92.41±0.18 |
| | DELTA [19] | 87.17±0.23 | 89.95±0.25 | 91.17±0.19 | 92.29±0.12 |
| | **DELTA+BSS** | 87.30±0.23 | 90.44±0.12 | 91.70±0.30 | 92.62±0.27 |
| FGVC Aircraft | $L^2$ | 39.57±0.20 | 57.46±0.12 | 67.93±0.28 | 81.13±0.21 |
| | **$L^2$+BSS** | 40.41±0.12 | 59.23±0.31 | 69.19±0.13 | **81.48**±0.18 |
| | $L^2$-SP [20] | 39.27±0.24 | 57.12±0.27 | 67.46±0.26 | 80.98±0.29 |
| | **$L^2$-SP+BSS** | 40.02±0.15 | 58.78±0.26 | 68.96±0.21 | 81.27±0.31 |
| | DELTA [19] | 42.16±0.21 | 58.60±0.29 | 68.51±0.25 | 80.44±0.20 |
| | **DELTA+BSS** | **43.79**±0.19 | **61.58**±0.17 | **69.46**±0.29 | 80.85±0.17 |

**Sensitivity Analysis.**  Sensitivity analysis of a larger $k$ in Equation (8) is conducted on the Stanford Dogs dataset, with results shown in Figure 3(d). When the amount of training data examples is small, a larger $k$ enhances the performance of fine-tuned models. However, with relatively sufficient training data examples, a larger $k$ leads to a slight decline in classification accuracy. Thus $k = 1$ is generally a good choice, since we always have difficulty in determining the relative size of the training dataset.

## 5.3   More Scenarios

**Incremental Learning.**  The fine-tuning step is a special case of incremental learning that has only one additional stage. Though source task is not considered by BSS, it is interesting to find how BSS influences the performance of incremental learning methods. We evaluate BSS embeded with **EWC** [14] on the permuted MNIST dataset. For this task, we use the same training strategies of [14] and test the accuracy of both the source task and target task. Top-1 classification accuracies are shown in Table 3. It is observed that BSS promotes the target task while slightly hurts the source task. This is an intuitive and reasonable result because BSS tries to alleviate the risk of negative transfer and does not focus on remembering the previously-learnt knowledge of the source task.

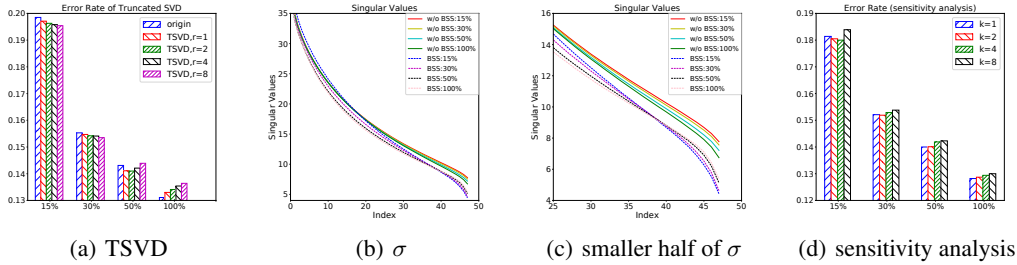

| (a) TSVD | (b) $\sigma$ | (c) smaller half of $\sigma$ | (d) sensitivity analysis |

Figure 3: Analysis of TSVD, singular values and hyperparameter sensitivity: (a) Error rate of TSVD with different $r$; (b) All singular values of feature matrices in four configurations for Stanford Dogs, which have random sampling rates 15%, 30%, 50% and 100% respectively, either with (w/) BSS and without (w/o) BSS; (c) Smaller half of singular values in (b); (d) Sensitivity analysis of different $k$.

Table 3: BSS Embedded into EWC for Incremental Learning

| Method (incremental learning) | task A | task B | Avg |
|---|---|---|---|
| fine-tuning + EWC [14] | 96.60 | 97.42 | 97.01 |
| fine-tuning + EWC [14] + **BSS** | 96.46 | 98.04 | 97.25 |

Table 4: BSS Embedded into BERT for Nature Language Processing

| Method (text classification) | MNLI-m | QNLI | MRPC | SST-2 | Avg |
|---|---|---|---|---|---|
| $\text{BERT}_{\text{base}}$ [6] | 84.4 | 88.4 | 86.7 | 92.7 | 88.0 |
| $\text{BERT}_{\text{base}}$ [6] + **BSS** | 85.0 | 89.6 | 87.9 | 93.2 | 88.9 |

**Natural Language Processing.** Fine-tuning is an important technique to transfer knowledge from other sources or pre-trained models. Its effectiveness in visual recognition applications is shown in section 5.2. We further justify its power in natural language processing. The General Language Understanding Evaluation (GLUE) benchmark [39] is a collection of diverse natural language understanding tasks. MNLI-m, QNLI, MRPC and SST-2 in GLUE [39] are used to evaluate the effect of BSS. Considering that BERT [6] is a state-of-the-art NLP pre-trained model, we embed BSS into $\text{BERT}_{\text{base}}$. We use a batch size of 32 and fine-tune for 3 epochs over the data for these four tasks. For learning rate, we use the same strategies as [6]. Results on four tasks in the Dev sets are listed in Table 4. From the Table we find that BSS can also help fine-tuning in natural language processing.

# 6 Conclusion

In this paper, we studied fine-tuning of deep models pre-trained on source tasks to substantially different target tasks. We delved into this widely-successful inductive transfer learning scenario from a new perspective: negative transfer. While existing deep methods mainly focus on alleviating the problem of catastrophic forgetting for reusing pre-trained knowledge, we find that not all weight parameters or feature matrices are transferable and some spectral components in them are detrimental to the target tasks, especially with limited training data. Based on this observation, Batch Spectral Shrinkage (BSS), a regularization approach based on spectral analysis of feature representations, is proposed to actively inhibit untransferable spectral components. BSS is pluggable into existing fine-tuning methods and yields significant performance gains. We expect that BSS will shed light into potential future directions for safe transfer learning towards making inductive transfer never hurt.

# Acknowledgments

We thank Dr. Yuchen Zhang at Tsinghua University for helpful discussions. This work was supported by the National Key R&D Program of China (2017YFC1502003) and Natural Science Foundation of China (61772299 and 71690231).

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
