[Reviews · NeurIPS 2019]

Reviewer 1



The method proposed in this paper are novel and the experiments are very solid. In table 2 they show the result evaluated in 5 datasets, 4 different sample rates and 3 different methods combined with the new method. The proposed method performs very well in all situations. However, I am not convinced by the explanation given in the paper. First of all, I don't understand what does transfer in 'negative transfer' means? Is it mean to fine tune from a pretrained model? Or to use the feature from a pretrained model? Or to use some technique like L2-SP to preserve the previous feature? The authors conclude that the negative transfer does exist in 3.2 and Fig 1(a). But from my prospective, fig 1(a) only shows that L2-SP performs worse than L2. Fig 1(b) shows that lower layers are more transferable than higher layers and the transferability can be somehow indicated by the singular value. Fig 1(c)(d) shows that with more training data, the singular values are smaller, especially in the smaller half of sigma. These are all very interest result, but I cannot see how it related to negative transfer.

Reviewer 2



This paper investigates whether negative transfer exists in fine-tuning deep networks pre-trained on a domain and how to deal with it. Batch Spectral Shrinkage (BSS) is proposed as a regularization to suppress which features that potentially cause negative transfer, indicated by small singular values of SVD(F), where F are the feature matrices. The main problem addressed by the paper is vivid and well motivated. The proposed regularization to alleviate the problem is intuitive, through the familiar singular value decomposition. The empirical evaluations show the effectiveness of the BSS regularization over a range of datasets -- models with BSS perform on par or better than those without BSS after fine-tuning, especially with limited number of fine-tuning examples. Improvements: - In my opinion, the fine-tuning step is a special case of continual learning (CL) that only has 1 additional step. It would be interesting if BSS can be incorporated into existing CL methods such as EWC (Kirkpatrick et al. 2017) and/or LwF (Li & Hoiem 2016) as well -- is it even possible to do so? - It would be great if there is a BSS fine-tuning use case other than visual recognition, e.g., text classification with pre-trained word embeddings, that can be evaluated. ============== After the rebuttal I thank the authors for providing the response to my concern, by reporting the additional experiment outcomes with EWC and on text classification. My final score is up to 1 level. Please incorporate the new experiment results into the manuscript / supplemental materials.

Reviewer 3



1. The reviewer thinks that the novelty of this paper is not enough. The title of this paper is “Catastrophic Forgetting Meets Negative Transfer”. However, the part that deals with catastrophic forgetting only uses the previous methods, and the formula only extends the proposed BSS regularization to the previous methods. There are also no ablation studies to verify the effectiveness of the two parts, i.e., catastrophic forgetting part and negative transfer part. 2. Line 167 mentioned that “in the higher layers, only eigenvectors corresponding to relatively larger singular values produce small relative angles. So aligning all weight parameters indiscriminately to the initial pre-trained values is risky to negative transfer.” Then why not re-initialize all the high-level parameters and train again? Part of the transfer learning, i.e., “A Survey on Transfer Learning”, only transfer the parameters of lower layers. Are there any experiments to verify the pros and cons of this process? 3. The paper analyzes the influence of network parameters and feature output representation on negative transfer. Why use feature regularization instead of parameter regularization? Are there any experiments to verify? 4. The paper mainly solves the negative transfer phenomenon in fine-tuning. But the comparison methods are all about catastrophic forgetting, and there is no negative transfer method. Why not compare with the state of the art negative transfer methods? “Characterizing and avoiding negative transfer” 2018; “Deep coral: Correlation alignment for deep domain adaptation” 2016 “Adapting visual category models to new domains” ECCV 2010 “Adversarial discriminative domain adaptation” CVPR 2017 5. Some statements in the paper are repeated, and the format of the reference is very confusing.

[Author Response · NeurIPS 2019]

We thank the reviewers for insightful comments. We have provided **code** in the supplemental for full reproducibility.

**Common Question: The method is for negative transfer rather than catastrophic forgetting.**

In transfer learning [23], "transfer" is the ability to apply knowledge learned in previous tasks to new tasks. Due to large
domain gap, only part of the pre-learned knowledge is useful for a new task: If the useful part is erased during transfer,
it is **catastrophic forgetting**; If the harmful part is preserved during transfer, it is **negative transfer** (Line 31-32).

Hence, catastrophic forgetting and negative transfer constitute a **dilemma** and should be mitigated jointly for optimal
performance. This is emphasized by the current Title and Introduction. While catastrophic forgetting has been studied
extensively by the community [34, 18], there has no work on mitigating negative transfer in fine-tuning. We propose a
novel approach to negative transfer, which is pluggable in the methods for catastrophic forgetting to tackle the dilemma.

**R1.1: How is Fig 1 (a)–(d) related to negative transfer? Why the new method works?**

**Fig 1(a)** shows that $L^2$-SP performs worse than standard fine-tuning $L^2$. This is a case of *negative transfer* by definition
[23]. In **Fig 1(b)** we delve into why negative transfer happens: the *Relative Angles* in the higher layers reveal that the
eigenvectors with smaller singular values are **not** transferable. This harmful part causes negative transfer in $L^2$-SP since
it preserves all pre-trained knowledge. Similar results are observed for DELTA [18] (will be added to complete Fig 1).

As justified by [2], with sufficient labeled data, fine-tuning and training from scratch achieve comparably best results—
negative transfer does not happen in this case. Hence in **Fig 1(c)–(d)**, we analyze the singular values in this case, and
find that the smaller singular values are suppressed more. This hints us that the knowledge conveyed by eigenvectors
with smaller singular values are the causes of negative transfer and should be *shrunk*. This well motivates our approach.

**R2.1: BSS with continual learning & BSS in text classification with pre-trained word embeddings.**

In the table below: **For continual learning**, we evaluate BSS with **EWC** [13] on the permuted MNIST dataset. BSS
promotes the target task while slightly hurts the source. **For text classification**, BSS enhances the performance of
**BERT** [1], a state-of-the-art NLP pre-trained model. Results on Dev sets are listed, with all hyper-parameters consistent.

| Method (continual learning) | task A | task B | Avg | Method (text classification) | MNLI-m | QNLI | MRPC | SST-2 |
|---|---|---|---|---|---|---|---|---|
| fine-tuning + EWC | 96.60 | 97.42 | 97.01 | $BERT_{base}$ | 84.4 | 88.4 | 86.7 | 92.7 |
| fine-tuning + EWC + BSS | 96.46 | 98.04 | 97.25 | $BERT_{base}$ + BSS | 85.0 | 89.6 | 87.9 | 93.2 |

**R3.1: Concern on novelty & compare with negative transfer methods in Domain Adaptation.**

Orthogonal to catastrophic forgetting, negative transfer is the bottleneck of transfer learning [23] and remains an *open*
*problem* in inductive transfer learning (a.k.a. fine-tuning in the context of deep learning). This work provides the first
approach to this important open problem, making a major contribution to this field.

Even in domain adaptation, there lacks in-depth analysis on negative transfer until [32] (CVPR'19). However, domain
adaptation and inductive transfer (fine-tuning) are completely different scenarios, detailed in the following table (left).
We are the first to address negative transfer in **fine-tuning**, to which domain adaptation methods cannot be applied.

The papers Reviewer #3 lists are important in domain adaptation. We will cite and discuss them in a future version.

| Method | labeled source samples | source labels vs. target labels | Method | 15% | 30% | 50% | 100% |
|---|---|---|---|---|---|---|---|
| fine-tuning | unavailable | different | $L^2$ | 73.95±0.18 | 79.43±0.23 | 81.40±0.21 | 84.77±0.32 |
| domain adaptation | available | identical | $L^2$ + re-initialize | 70.32±0.32 | 76.36±0.29 | 79.98±0.28 | 83.35±0.33 |

**R3.2: Why not re-initialize all the high-level parameters and train again?**

Fig 1 reveals that the eigenvectors with larger singular values in higher layers are transferable. If we re-initialize those
parameters, all pre-trained knowledge is discarded and *catastrophic forgetting* happens. Results on Stanford Dogs by
re-initializing Layer 4 in ResNet-50 are shown in the above table (right), which are worse than vanilla fine-tuning ($L^2$).

**R3.3: Why use feature regularization instead of parameter regularization?**

Parameter regularization has several disadvantages: **(1)** It is hard to decide weights of which layers should be regularized
(Line 191-192). In contrast, feature regularization can regularize each layer by taking the advantage of back-propagation.
**(2)** The parameters form high-dimensional matrix, whose SVD incurs unacceptable computational cost (Line 207-211).
In contrast, we can perform SVD over the feature matrix of each mini-batch, which only adds slightly more computation.

**R3.4: Ablation studies of the two parts (catastrophic forgetting & negative transfer).**

The ablation studies as requested by the reviewer have already been shown detailedly in Table 2: $L^2$ denotes the standard
fine-tuning; $L^2$ + BSS is the ablation study for the negative transfer part; $L^2$-SP / DELTA is the ablation study for the
catastrophic forgetting part; And $L^2$-SP + BSS / DELTA + BSS unifies the two parts to tackle the dilemma in them.

Through above it has been apparent that major questions raised by Reviewer #3 have been answered in the original paper.

**References**

[1] J. Devlin, M.-W. Chang, K. Lee, and K. Toutanova. Bert: Pre-training of deep bidirectional transformers for language
understanding. In *NAACL*, 2019.
[2] K. He, R. Girshick, and P. Dollár. Rethinking imagenet pre-training. *arXiv preprint arXiv:1811.08883*, 2018.


[Meta-Review · NeurIPS 2019]

The reviews as well as the rebuttal have generated interesting discussions about the aspects of transfer learning and domain adaptation discussed in this paper. Although there is not a clear consensus (one reviewer is oscillating between a weak reject and a weak accept), I found both the paper and the comments of the skeptical reviewer (Reviewer 3) were relevant. Thus, I believe that this contribution is worth presenting at the conference since it can inspire significant further developments.